# Calibration and assessment of electrochemical low-cost sensors in remote alpine harsh environments

Federico Dallo[1,2], Daniele Zannoni[3], Jacopo Gabrieli[1], Paolo Cristofanelli[4], Francescopiero Calzolari[4], Fabrizio de Blasi[1], Andrea Spolaor[1], Dario Battistel[2], Rachele Lodi[1], Warren Raymond Lee Cairns[1], Ann Mari Fjæraa[5], Paolo Bonasoni[4], and Carlo Barbante[1,2]

[1]Institute of Polar Sciences (CNR-ISP), Via Torino, 155, 30172, Venice, ITALY
[2]University Ca' Foscari of Venice, Dorsoduro 3246, 30123, Venice, ITALY
[3]Geophysical Institute, University of Bergen and Bjerknes Centre for Climate Research, Bergen, NORWAY
[4]Institute of Atmospheric Sciences and Climate (CNR-ISAC), Via P. Gobetti 101, 40129 Bologna, ITALY
[5]Norwegian Institute for Air Research (NILU), Instituttveien 18, 2007 Kjeller, NORWAY

**Correspondence:** Federico Dallo (federico.dallo@unive.it)

**Abstract.**

The present work presents the results obtained using an original open-source low-cost sensor (LCS) system developed to measure tropospheric $O_3$ in a remote high altitude alpine site. We conducted our study at the Col Margherita Observatory (2543 $\mathrm{m}$ a.s.l.), a World Meteorological Organization Global Atmosphere Watch Regional Station, located in the Italian Eastern Alps. The sensing system mounts three equivalent commercial low-cost sensors that have been calibrated using a laboratory standard (Thermo 49iPS), referenced to the Standard Reference Photometer #15 calibration scale by the WMO, before field deployment. Intra and inter-comparison between sensors and reference (Thermo 49c) have been conducted for seven months from May to December 2018. The sensors required an individual calibration, both in laboratory and in the field. The sensor's dependence on the environmental meteorological variables has been considered and discussed. The evaluation of the analytical performances of this sensing system provides an $LOD < 5$ ppb, $LOQ < 17$ ppb, $LDR$ up to $250$ ppb, intra-Pearson correlation coefficient (PCC) up to $0.96$, inter-PCC $> 0.8$, bias $> 3.5$ ppb and $\pm 8.5$ at 95% of confidence. Thanks to the first implementation of an LCS System in an alpine site, it has been demonstrated how it is possible to obtain valuable data from a low-cost instrument in a remote environment. This opens new perspectives for the adoption of a low-cost sensor network in atmospheric sciences.

## 1 Introduction

The troposphere is a very complex system, which is subject to continuous inputs, production and removal processes of ozone from natural phenomena and human activities (lifetime 25 days(Young et al., 2013)). In southern Europe the background tropospheric ozone concentration appears to be significantly affected by three main air mass transport processes: (i) transport of polluted air masses on regional and long-range scales; (ii) downward transport of stratospheric air masses; (iii) transport of mineral dust(Cristofanelli and Bonasoni, 2009). Large gaps remain in the surface observation network, despite many years of research and monitoring of surface ozone on regional and global scales. Especially in terms of areas without monitoring and in terms of regions that have monitoring programs but no public access to the data archive(Schultz et al., 2017). Future improvements in the database would require better data harmonization, enhanced data sharing and monitoring in data-sparse regions to develop and integrate in-situ networks, complementary to satellite instruments, in order to improve measurement accuracy and spatial/temporal sampling(O'Neill et al., 2015). Therefore covering in-situ spatial data gaps to increase the effectiveness of satellite observations, which must be calibrated using ground-base reference measurements(Organization, 2017), is necessary to achieve a better agreement between observations and models.

Tropospheric ozone was chosen for this pilot study due to its high relevance to the Earth's climate(Tørseth et al., 2012), ecosystems and human health. It is one of the most important atmospheric gases involved in photochemical reactions(Crutzen et al., 1999). Ozone is the precursor of oxidizing substances like $OH^-$ and $NO_3^-$ and it is a key agent determining the oxidation capacity of the troposphere(Gauss et al., 2003). Tropospheric ozone influences climate as it plays a central role in the radiative budget of the atmosphere (IPCC, 2013 p.55) and it is the third most important greenhouse gas in the free troposphere(Forster et al., 2007). Furthermore, surface ozone is a dangerous secondary pollutant causing harm to human health and ecosystems(Cooper et al., 2014; Jacobson and Jacobson, 2002).

Earth monitoring is a key aspect in improving our understanding of global processes and Climate. In this framework, remote areas, such as mountain regions, are considered reference background sites due to their sensitivity to Climate Change(Bonasoni et al., 2008; ISAC-CNR; Cristofanelli et al., 2006; Barbante et al., 2004; Gabrieli and Barbante, 2014). Therefore data coming from high-altitude observatories provide valuable insights on the Earth's Climate and are crucial for the climate communities.

At present, the problem of establishing the spatio-temporal representativeness of measurements of ozone remains a difficult task, especially in presence of great spatial variability as in the case of remote regions. Increasing the number of reference-grade observatories devoted to long-term baseline observations in the alpine area is not practicable due to the high costs of construction, maintenance and labour. Moreover, global atmospheric observatories have to be located in remote areas to reduce the influence of local source pollution, thus increasing the logistical costs and discomfort for the personnel. In this context, remote-sensing can not fully solve the spatial problem as satellite systems can only meet the established requirements if they are supported by correlative data of known quality, reliable ground-based observations and quantitative science(Dobber et al., 2006; ESA, 2020).

Emerging commercial Low-Cost Sensors (LCSs) present an unique opportunity to overcome the challenge of increasing the spatial density of monitoring sites. The rapid development and continuous improvement of low-cost technologies are

demonstrating notable applications(Hertel et al., 2015; Hagan et al., 2018). Nowadays high-quality LCSs are beginning to play a role in areas such as modelling or emissions validation in support of state-of-the-art instrumentation and established networks(Mead et al., 2013; Win; Borrego et al., 2016, 2018; Heimann et al., 2015; Castell et al., 2015). While most Low-Cost Sensor Network applications are designed for the built environment(Mueller et al., 2017; Jiang et al., 2005; Bauman et al., 2013; Andersen and Culler, 2014; Levis et al., 2005; Andersen et al., 2017) (e.g. smart cities, indoor air quality), there is a lack of studies in remote alpine regions where data are crucial for the Climate and meteorological research community. The World Meteorological Organization Global Atmosphere Watch (WMO-GAW) recognizes that the fate of the next generation of monitoring stations could be dramatically modified by the breakthroughs of new LCSs technologies(Lewis et al., 2018), but there are still open issues to be addressed such as: (i) standardization of tools and protocols to ensure growth in the number of low-cost nodes without having to fundamentally change the LCSs network architecture, (ii) compatibility with established observing systems architecture (e.g. Zhang and Director (2010)), (iii) the simplification of the remote system management and (iv) the assurance of data quality.

In this work we aim to assess the reliability of LCSs for monitoring near-surface ozone concentrations in a remote Alpine region, focusing on the *precision*, *accuracy* and *reliability* of LCSs measurements compared with a reference instrument. We carried out two laboratory calibration experiments, in April and July 2018, in order to evaluate the LCSs performances and their stability over time in a controlled environment before field deployment. We conducted our study from March to December 2018 at the Col Margherita Atmospheric Observatory (MRG). Due to the harsh weather conditions recorded at Col Margherita, this site was considered ideal to test the performances of the LCSs in view of the modern challenge of deploying low-cost applications in real world difficult situations.

## 2 Materials and Methods

### 2.1 Site description

Col Margherita site is characterized by an alpine climate. Considering the 2008 - 2018 time window, the annual mean temperature was 3.2 °C, with August as the warmest month with an average temperature of 11.8 °C and January as the colder month with an average temperature of -4.7 °C. The average annual rain precipitation was 1485 mm, with August as the wettest month with 161.9 mm and December as the driest month with 59.5 mm(Trentino). Although not particularly high, the location is representative of the synoptic conditions of the free troposphere as represented on maps at around 700 hPa. This is possible since the site is distant and scarcely influenced, by surrounding orographic barriers (Figure 1). Despite the location could have been affected by pollutants emitted by major cities and transported by local winds(Masiol et al., 2015; Diémoz et al., 2019), recent studies showed that the atmospheric composition of Col Margherita is related to air masses on a regional scale; in particular during winter the observatory is located above the atmospheric boundary layer so that local sources are not significant(Barbaro et al., 2020; Sprovieri et al., 2016).

About 20 thousand people live within 10 km of the MRG observatory area and about 70 thousand within 50 km(ISTAT, 2011; Veneto, 2020; gis, b, a).

### 2.2 MRG Observatory description

The MRG observatory is a GAW Regional Station (WIGOS Id: 0-380-0-MRG) led by the Institute of Polar Sciences of the National Research Council of Italy (ISP-CNR). It is located on the southern-eastern side of the Italian Alps (46.36683 N, 11.79192 E), based at an altitude of 2543 meters above mean sea level (m a.s.l.), it was chosen as representative for the surrounding alpine region(Barbaro et al., 2020).

The observatory is a prefabricated insulated shelter with external dimensions of 3.00m x 2.42m x 3.22m. The observatory is equipped with a complete automatic weather station mounted on a 3 m mast (AWS: CS215-L probe, Campbell Scientific, Utah, USA; PTB110 Barometer, Vaisala, Helsinki, Finland; 05103-45 Wind Monitor, R. M. Young Company, Michigan, USA) and an UV-absorption ozone analyser (Thermo Fisher Scientific 49C, SN: 0503110399). The MRG observatory is unmanned and fully automated, it is connected to the main electrical grid and supplied with a backup solar power system with ≈200 Ah batteries in case of grid failure. The observatory is equipped with remote control via GSM / GPRS technology.

### 2.3 Low-cost sensors

We used the Alphasense OX-B431 commercial passive sensor which belongs to the class of electrochemical sensors that operate in amperometric mode. These oxidising sensors generate a current that is linearly proportional to the fractional volume of the target gases. The gas sensor provides three electrodes: the working electrode (WE) where the ozone reacts and a current proportional to the ozone concentration is formed; the counter electrode (AUX) that balances the reaction of the working

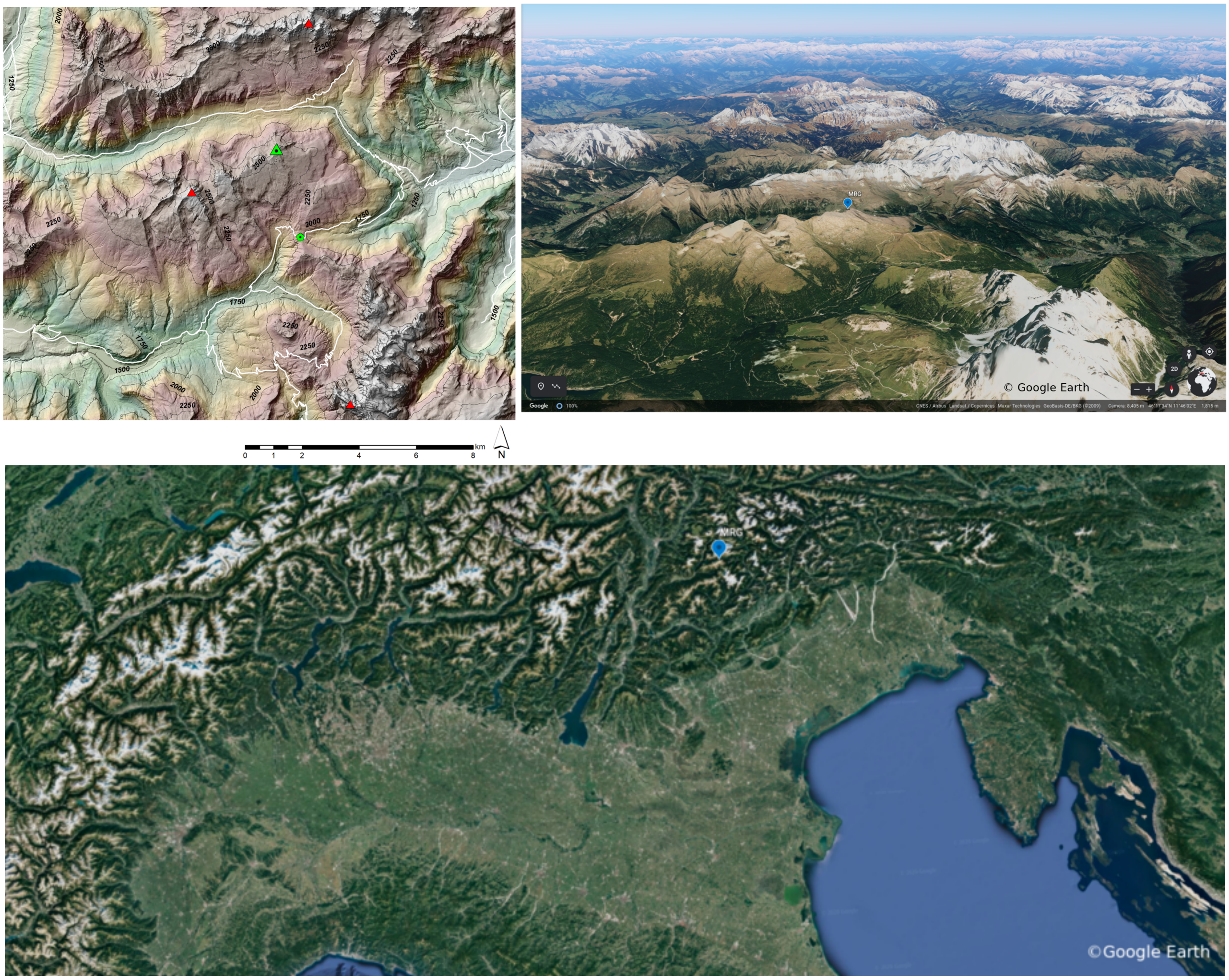

**Figure 1.** From top left to bottom. (a) Surrounding area of the Col Margherita Observatory (MRG, 46.36683 N, 11.79192 E, 2543 m a.s.l.). Geographical key points and their distance from the MRG observatory are: Passo Valles (SSE) at 2032 m a.s.l. and 3.2 km away, Cima Bocche (ESE) at 2745 m a.s.l. and 3.3 km away, Cima dell'Uomo (NNE) at 3010 m a.s.l. and 4.6 km away and Cimon della Pala (SSE) at 3184 m a.s.l. and 9.3 km away. (b) 3D aerial view of the Col Margherita (from © Google Earth). (c) Satellite view of the north-east of Italy and the location of the Col Margherita Observatory (from © Google Earth).
electrode to generate an equivalent current in the opposite direction and the reference electrode that anchors the working electrode potential to ensure that it is maintained at a fixed potential, independently from the current generated.

Each sensor was purchased with its individual sensor board that provides a dual channel voltage output, one for the working electrode and one for the auxiliary electrode. These output pins can be interfaced with a commercial Analog to Digital Converter.

Alphasense provides a calibration certificate and reference regression coefficients for each sensor. The manufacturer declares that the calibration was conducted in a controlled environment (temperature 22 $\pm$ 3 °C, relative humidity 40 $\pm$ 15 %) using the ozone generator - Thermo Scientific Model 49i-PS (flow 0.5 $l \cdot min^{-1}$) and it was performed considering two points, zero and span (1 part per million (ppm)). The technical sheet (www.alphasense.com) reports that the sensor limit of detection is in the range of units of parts per billion (ppb), a sensitivity that is required for detecting environmental ozone concentration. The manufacturer declares that the sensor voltage output is linear up to 20 ppm of the target gas.

Three equivalent sensors were installed on the sensing system in order to evaluate intra-compatibility between measurements and calibration stability as a function of time. Laboratory calibration was performed at the CNR-ISAC headquarters (Bologna, Italy) before field deployment. Field calibration was evaluated through inter-comparison with a reference UV-absorption O3 analyser installed at the monitoring site. The reference instrument was referred to the Standard Reference Photometer #15 (SRP-15) calibration scale(GAW Report No. 252 WCC-Empa Report No. 19/3) through an intercomparison with the calibrator hosted at the CNR-ISAC laboratory (s/n: 1404860524) in 2017, June.

### 2.4 Sensor node

The sensing system (Figure 2) was designed to be easily built. It consists of parts that may be purchased online and they are easy to replace in case of failure. The sensing system was composed of an IP56 enclosure with three holes for the working low-cost sensors and two additional holes for waterproof power and Ethernet connectors. An additional LCS was placed inside the box as a spare sensor. It was not necessary to use it throughout the experiment. The enclosure was neither heated nor regulated nor insulated. Sensor's holes, located at the bottom side of the enclosure, were watertight with rubber O-rings seals. A bag of silica gel was placed inside the box to keep the environment dry.

The sensing system was connected to the main power grid (AC) and to the LTE router of the MRG Observatory to allow data communication and remote control. The sensing system was equipped with an uninterruptible power supply (lead acid battery and battery charge regulator) in case of AC failure. The processing unit of the sensing system was a Raspberry Pi 3b+ and the analog signal of the sensors were digitized using a $\pm$ 5V (16 bit) ADC converter (Texas Instruments ADS1115). More detailed description of the hardware components of the sensing system and the approximate cost of the system's part are reported in Supplementary Materials S1.

### 2.5 Sensors calibration

Before field installation, LCSs were calibrated by comparing the analog voltage output of the low-cost sensors with the ozone concentration generated by an ozone calibrator (thermo scientific 49i-PS) with traceability to the WCC-Empa SRP#15 (cal-

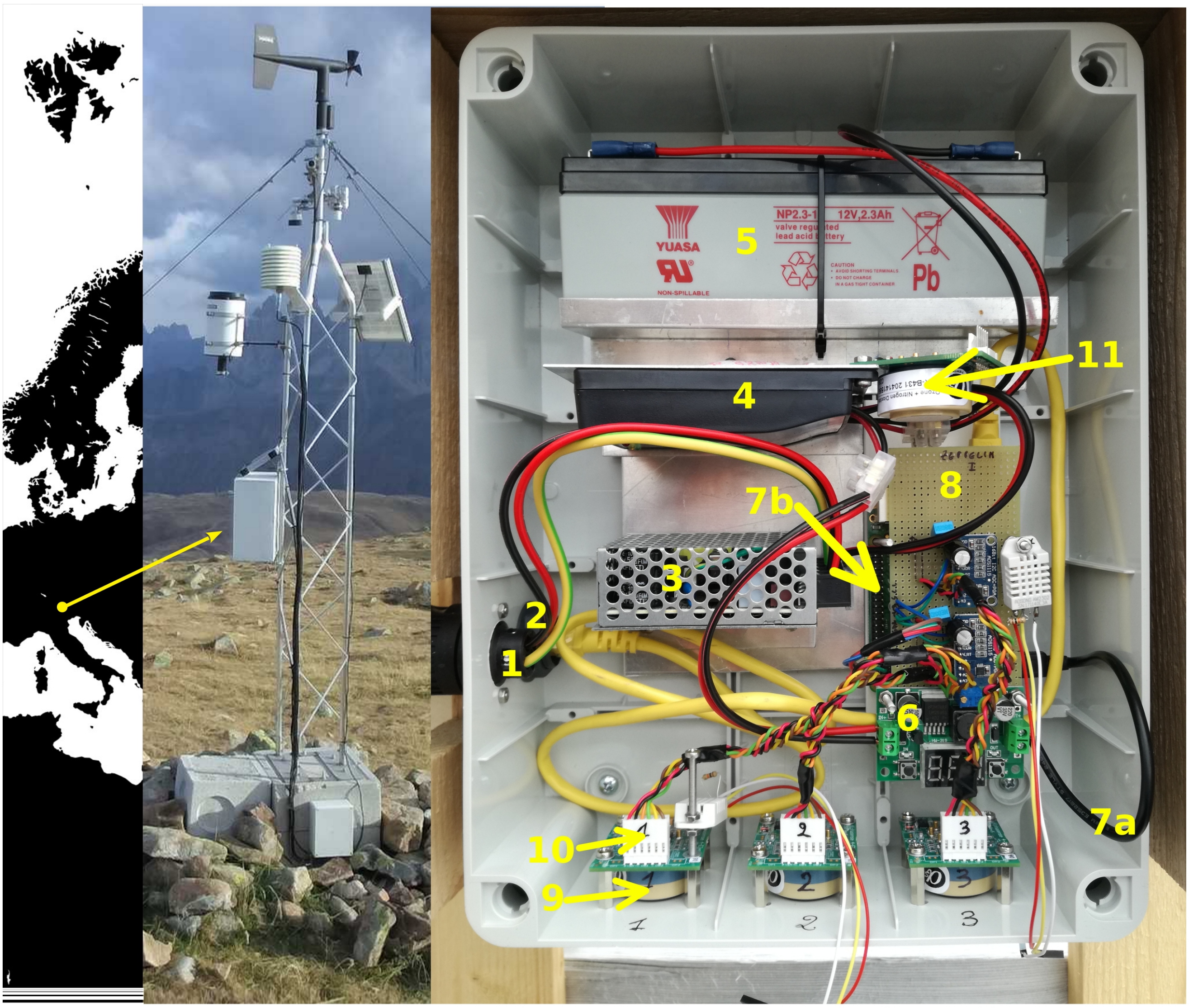

**Figure 2.** The low-cost sensing system and its components. 1) AC power IN, 2) Ethernet, 3) Power Supply, 4) Charge regulator, 5) Battery, 6) DC/DC converter, 7a) USB power wire, 7b) Raspberry Pi, 8) Data acquisition board, 9) Working sensor, 10) Sensor's plug, 11) Spare sensor.

ibration in 2017). Each sensor was installed on a specific gassing hood equipped with 1/4" barbed Swagelok fitting through which the calibration gas was fluxed using a digital mass flow meter. The calibration procedure was designed for the calibration of reference-grade instruments used in the WMO network. It consists of 21 steps in which the reference ozone (flow 0.9 $\mathrm{l \cdot min^{-1}}$) span from 0 to 250 $\mathrm{ppb}$ with a declared precision of 1 $\mathrm{ppb}$ and zero noise of 0.25 $\mathrm{ppb}$. Each step lasts for 20 minutes and the sensor total calibration time is 7 $\mathrm{hr}$. The reference concentration step sequence simulates a pseudo-random variation of ozone concentration. This procedure allows to evaluate the sensor's tolerance, ability to respond to sudden and unpredictable (small/large, increasing/decreasing) changes in the ozone concentration, to assess whether the sensor suffers from memory effect or loss of sensitivity and to estimate drifts of the instrumental response. The reproducibility and the stability of the sensors calibration parameters were evaluated performing two laboratory calibration. Three sensors (Mrg1: Alphasense Serial Number 204141855; Mrg2: S/N 204141543; Mrg3: S/N 204141544) were calibrated in Bologna between the $18^{th}$ and the $19^{th}$ of April 2018, deployed at Col Margherita for a month, then removed the $22^{nd}$ of June and re-calibrated between the $11^{th}$ and the $12^{th}$ of July under the same conditions of the first calibration. The sensing system was finally re-installed at the Col Margherita Observatory on the $17^{th}$ of July. During both calibration period, laboratory temperature and humidity were controlled and maintained at ≈22.5 $^{\circ}\mathrm{C}$ and ≈50$\%$ respectively. During laboratory calibration the analog voltage output of the LCSs was recorded every second. Values were then aggregated to 1-minute averages and time-matched with the ozone concentration dataset generated (1-minute) by the reference instrument.

### 2.6 Field installation and data acquisition parameters

The installation of the sensing system was carried out the $25^{th}$ of May 2018 at the Col Margherita Observatory. The LCSs instrument was mounted in the AWS mast, 1 $\mathrm{m}$ above the ground. The gas inlet through which the reference UV-absorption ozone analyser sucks in the air ($\approx 0.6\ \mathrm{l \cdot min^{-1}}$) was located on the roof of the observatory at 5 $\mathrm{m}$ above the ground. The comparison between the reference instrument in Col Margherita and the LCSs of ozone was performed when both the systems were running, considering the time window from May 2018 to December 2018. The analog voltage output of the LCSs was recorded every 5 seconds during the field experiment. The values were treated for the data processing as described in Sect. 2.7, then aggregated to hourly averages and finally time-matched with the reference ozone concentration dataset validated through manual and automatic data preprocessing(Naitza et al., 2020).

### 2.7 LCSs data processing

The LCSs raw data were preprocessed to discover possible outliers during the laboratory calibration and the field evaluation. The filter used was based on the computation of a local polynomial (R function LOESS, R Core Team; Cleveland et al. (1992)) and the median absolute deviation (MAD) between this polynomial and the measurements within a moving window. We define the outliers as measurements deviating more than 5 times the MAD from the local polynomial(Mueller et al., 2017). If the MAD was smaller than the 50% quantile of all differences (| local polynomial - measurement |) it was substituted by this value. This approach prevents the exclusion of measurements during time periods with almost no variation on the ozone concentration. We considered a time window of 20 seconds, chosen after evaluating the time series autocorrelation lag (R function ACF, R core

Team; Brockwell et al. (1991)) for the laboratory calibration LCS dataset. The resulting laboratory calibration outliers were less than the 0.1% of the total LCS measurements (see Supplementary Figure S2.1).

A time window of 1 hr was considered before averaging the data to hourly means for the field LCS dataset. This procedure excluded less than 0.5% of 5-second raw data, mainly generated during the turning on phase of the LCS system (see Supplementary Figure S2.2). Minutes containing less than 9 valid observations (75 %) were excluded and hourly averages were considered if there were data for at least 45 minutes (75 %). Besides, 82 hours for the Mrg2 sensor were manually excluded from the dataset. From 1AM of the $5^{th}$ of November to 11PM of the $8^{th}$ of November, the WE electrode of the Mrg2 sensor showed a completely different behavior compared to the other two sensors (see Supplementary Figure S2.3).

Harsh weather condition caused many periods of absence of main AC current at MRG observatory. This made the reference ozone dataset discontinuous during the summer. On September $14^{th}$, a problem on the pump of the reference instrument was diagnosed and the reference instrument was dismantled for maintenance until the $25^{th}$ of October. The $29^{th}$ of October we faced a power outage due to a severe storm ("Vaia", $29^{th}$ of October 2018, see Supplementary Figure S5.1). Another problem with the pump of the reference instrument was encountered the $14^{th}$ of December and the instrument was dismantled.

Therefore the final comparison between the reference instrument and the LCSs of ozone was performed on the ≈45% of the LCS's data, considering the time window from the $30^{th}$ of May 2018 to the $14^{th}$ of December 2018, when both the systems were running.

Evaluation of LCSs accuracy was considered skimming data if threshold value for relative humidity (RH) was overcome. We considered a multivariate ANOVA model to evaluate the effect of the meteorological variables on the LCSs measurements.

Limit of Detection (LOD) was calculated as the average zero signal plus three times the standard deviation. Limit of Quantitation (LOQ) was calculated as the average zero signal plus ten times the standard deviation(Committee et al., 1987; for Standardization, 1994; Harris, 2010).

## 3 Results

### 3.1 Laboratory calibration of LCSs

Laboratory calibration was performed through a linear model:

$$V_{OUT} = \beta_0 + \beta_1 \cdot [O_3] \quad (1)$$

where $V_{OUT} = V_{WE} - V_{AUX}$ is the LCS's analog output signal, $[O_3]$ is the ozone reference concentration in $\mathrm{ppb}$, $\beta_0$ (mV) is the intercept, $\beta_1$ $(\mathrm{mV} \cdot \mathrm{ppb}^{-1})$ is the slope. Linear model agreement between the reference and the LCS was evaluated using the Pearson correlation coefficient (PCC). Evaluation of bias was performed using the Mean Absolute Error (MAE):

$$MAE = \frac{\sum_{i=1}^{n} |e_i|}{n} \quad (2)$$

where $e_i$ was the difference between the prediction $(y_i)$ and the *true* value $(x_i$, $e_i = y_i - x_i)$ and *n* was the number of observations.

Summary of the calibration experiments and comparison with the calibration values declared by Alphasense are reported in Table 1. As an example of the laboratory experiment, the Mrg1 calibrated ozone measurements and analog output signal registered against the reference ozone in July are shown in Figure 3 using the main and secondary y-axis respectively. The linear regressions of the laboratory calibrations are shown in Figure 4. We did not see a change in the analytical performances of the LCSs between the two calibration experiments (Two-sample t-test, p-value > 0.7).

The mean voltage response of Mrg1 was 18.1 $\pm$ 0.6 $\mathrm{mV}$, when reference ozone concentration was 0.4 $\pm$ 0.1 $\mathrm{ppb}$, and reached 100.2 $\pm$ 0.6 $\mathrm{mV}$ when reference ozone concentration was 249.75 $\pm$ 0.04 $\mathrm{ppb}$. The precision of the Mrg1 sensor, calculated as the relative standard deviation (RSD), was $\approx$ 3.2% close to the LOD, and decreased to $\approx$ 0.6% for ozone concentrations higher than 200 $\mathrm{ppb}$ (Thompson (1988); Horwitz and Albert (1997), see Supplementary Figure S4.1). MAE for Mrg1 was 3.6 $\mathrm{ppb}$. LOD for Mrg1 was 4 $\mathrm{ppb}$. LOQ for Mrg1 was 14 $\mathrm{ppb}$. The instrumental response to ozone concentration was linear in the interval considered ($R^2$ = 0.998). Through our laboratory experiment we can confirm the linear dynamic range (LDR) for Mrg1 between the LOD (4 $\mathrm{ppb}$) and 250 $\mathrm{ppb}$. Alphasense datasheet reports that the instrumental response of the OX-B431 sensor was linear up to 2 $\mathrm{ppm}$.

The mean voltage response of Mrg2 was 0.8 $\pm$ 0.7 $\mathrm{mV}$, when reference ozone concentration was 0.4 $\pm$ 0.1 $\mathrm{ppb}$, and reached 89.3 $\pm$ 0.9 $\mathrm{mV}$ when reference ozone concentration was 250.2 $\pm$ 0.3 $\mathrm{ppb}$. Precision of the Mrg2 sensor was 3.5% for values close to the LOD and decreased to 0.9% for ozone concentrations higher than 200 $\mathrm{ppb}$. Mrg2's MAE was 2.5 $\mathrm{ppb}$, LOD was 5 $\mathrm{ppb}$, LOQ was 17 $\mathrm{ppb}$ and LDR was 5-250 $\mathrm{ppb}$ ($R^2$ = 0.999).

The mean voltage response of Mrg3 was 38.3 $\pm$ 0.9 $\mathrm{mV}$, when reference ozone concentration was 0.4 $\pm$ 0.1 $\mathrm{ppb}$, and reached 123.5 $\pm$ 0.9 $\mathrm{mV}$ when reference ozone concentration was 249.7 $\pm$ 0.2 $\mathrm{ppb}$. Precision of the Mrg3 sensor was 2.6% for values close to the LOD and decreased to 0.7% for ozone concentration higher than 200 $\mathrm{ppb}$. MAE for Mrg3 was 2.6 $\mathrm{ppb}$. Mrg3's LOD was 3 $\mathrm{ppb}$, LOQ was 9 $\mathrm{ppb}$ and LDR was 3-250 $\mathrm{ppb}$ ($R^2$ = 0.998).

Compared to the linear regression parameters given by the manufacturer, we obtained an average difference of about 4.2% on the intercept and an average difference of about 21.6% on the slope.

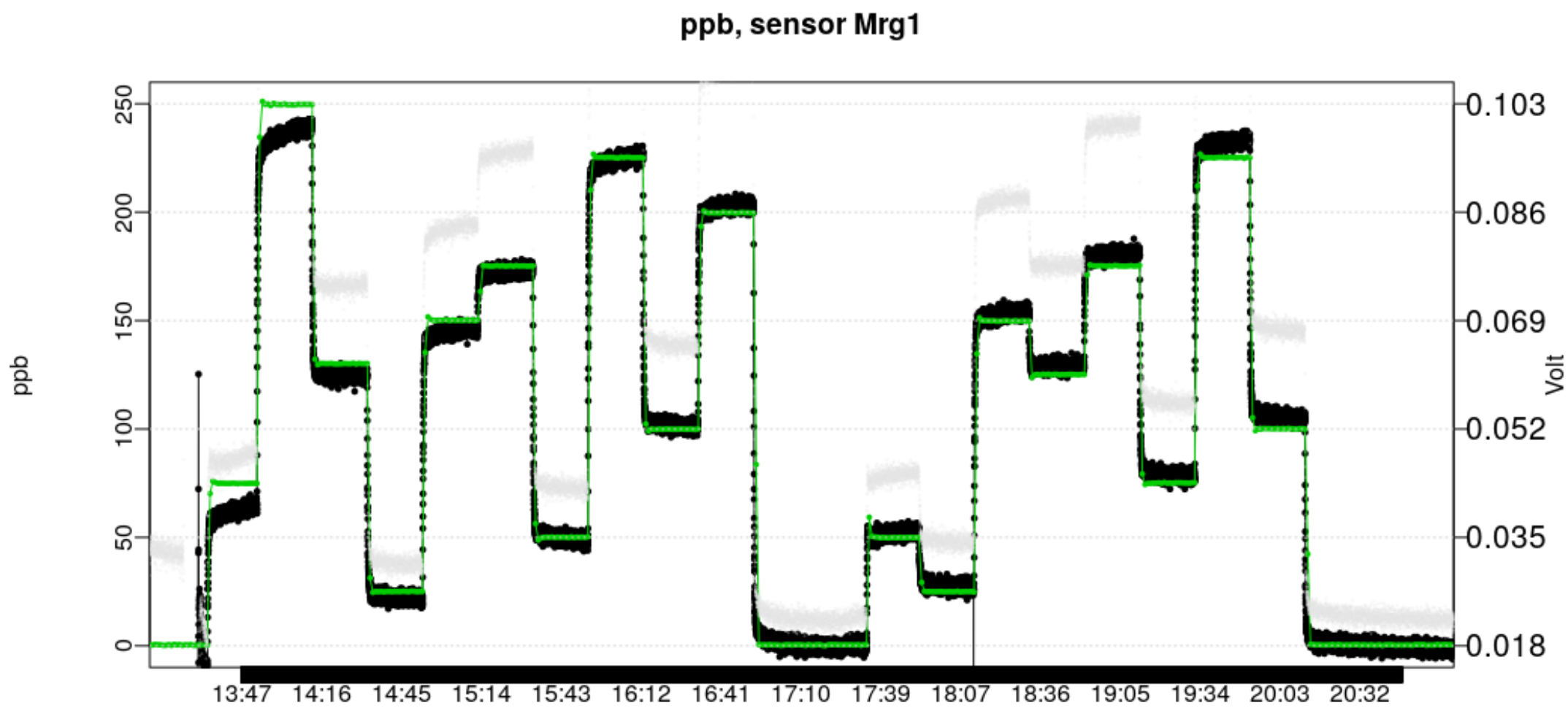

**Figure 3.** After calibration, voltage data ($V_{OUT}$) of the low-cost sensor is expressed in ppb (black) and compared to the reference ozone concentration (green). Sensor's ozone concentration obtained using the intercept and the regression coefficient provided by Alphasense are also shown (gray). Calibration date: 2018-07-11.

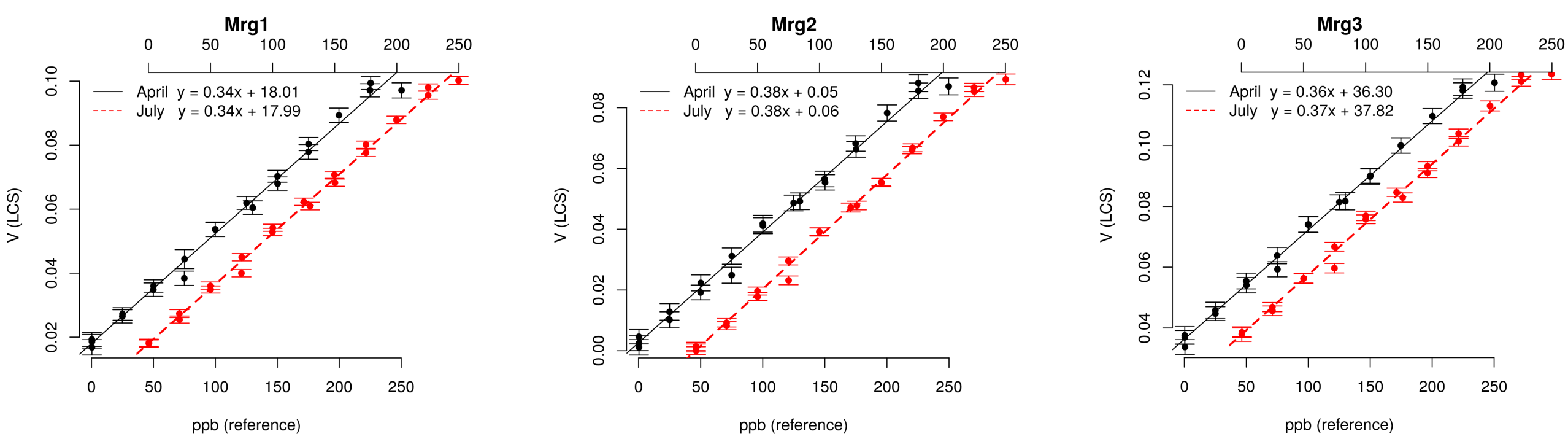

**Figure 4.** Laboratory calibration of the low-cost sensors. Linear regressions obtained during April (black) and July (red) are shown. From left to right are reported the results for sensors Mrg1, Mrg2 and Mrg3 respectively. To improve the visibility of the pictures two shifted x axis are given: the bottom axis refers to the April experiment while the upper axis refers to the July experiment.

| | Sensor | PCC | $\beta_0$ (mV) | $\beta_1$ ($mV \cdot ppb^{-1}$) | LDR (ppb) | RSD (%) | MAE (ppb) | LOD (ppb) | LOQ (ppb) | Calibration Date |
|---|---|---|---|---|---|---|---|---|---|---|
| Lab. Calibration | Mrg1 | 0.9958 | 18.01 | 0.34 | 4-225 | 3.2-0.6 | 3.6 | 4 | 14 | 2018-04-18 |
| | Mrg1 | 0.9982 | 17.99 | 0.34 | | | | | | 2018-07-11 |
| | Mrg2 | 0.9971 | 0.05 | 0.38 | 5-225 | 3.5-0.9 | 3.6 | 5 | 17 | 2018-04-18 |
| | Mrg2 | 0.9979 | 0.06 | 0.38 | | | | | | 2018-07-11 |
| | Mrg3 | 0.9977 | 36.30 | 0.36 | 3-225 | 2.6-0.7 | 2.6 | 3 | 9 | 2018-04-19 |
| | Mrg3 | 0.9970 | 37.82 | 0.37 | | | | | | 2018-07-12 |

| | Sensor | $MAE_{\beta_0',\beta_0}$ | $MAE_{\beta_1',\beta_1}$ | $\beta_0'$ (mV) | $\beta_1'$ ($mV \cdot ppb^{-1}$) | | | | | Calibration Date |
|---|---|---|---|---|---|---|---|---|---|---|
| Alphasense | Mrg1 | 1 | 0.07 | 17 | 0.27 | up to 2000 | - | - | - | - | 2018-02-06 |
| | Mrg2 | 2 | 0.10 | -2 | 0.28 | up to 2000 | - | - | - | - | 2018-02-06 |
| | Mrg3 | 2 | 0.07 | 36 | 0.30 | up to 2000 | - | - | - | - | 2018-02-06 |

**Table 1.** In the table above are reported the linear model regression coefficients and accuracy metrics obtained during the laboratory calibration. Pearson correlation coefficient (PCC), intercept ($\beta_0$), regression coefficient ($\beta_1$), Linear Dynamic Range (LDR), Relative Standard Deviation limits (RSD), Bias (MAE), Limit of Detection (LOD) and Limit of Quantitation (LOQ). In the table below are reported the regression coefficients, intercept ($\beta_0'$) and slope ($\beta_1'$) transcribed from sensor's datasheet. Mean Absolute Error (MAE) between $\beta_i'$ and $\beta_i$ (averages) is given in the first two column. LCSs statistics have been performed over 300 1-sec values for each ozone concentration step, for a total of $\approx$ 6.5k calibration points per sensor. Significant digits are in accordance with the calibration datasheet.

### 3.2 Field experiment

Field measurements were conducted from the $30^{th}$ of May 2018 to the $14^{th}$ of December 2018. We measured about 4800 hourly data that were collected in a wide range of environmental temperature (°C), pressure (hPa), wind speed ($\mathrm{m \cdot s^{-1}}$) and RH (%) as summarized in Table 2.

| | Temperature ($°C$) | Relative Humidity ($\%$) | Atmospheric Pressure ($hPa$) | Wind Speed ($m/s$) |
|---|---|---|---|---|
| min | -16.7 | 10 | 728.8 | 0.0 |
| median | 5.5 | 84.1 | 753.2 | 3.1 |
| mean | 4.0 | 79.1 | 753.6 | 3.8 |
| max | 18.3 | 100 | 771.7 | 34.9 |

**Table 2.** Meteorological data at Col Margherita evaluated from the $30^{th}$ of May 2018 to the $14^{th}$ of December 2018.

### 3.3 Correlation and bias between LCSs. Intra-comparison

A first estimate to verify that the LCSs were in agreement with each other during field measuring was done calculating their (PCC) and bias, referred as MAE, throughout the period considered, and their variation trends over time. A summary of the inter-PCC and inter-MAE trends are reported in (Table 3). The comparison is consistent except for the month of October. Out of the 744 hourly observations collected during the month, only 59 were considered for the analysis. From the $25^{th}$ to the $28^{th}$ of October, when both the LCS system and the reference instrument were working (see subsection 2.7). Since there were no signs of malfunctions and given the low environmental variability of the ozone concentration during the three days analysed (29 ppb < $O_3$ < 47 ppb calculated by the reference instrument and 25 ppb < $O_3$ < 46 ppb calculated averaging the LCSs measurements), we hypothesized that the low correlation value between LCSs could have been due to the inherent variability of the sensors' measurements. Indeed, if all the 744 LCSs hourly observations of October were considered, we could have calculated: $PCC_{1,2} = 0.86$,$PCC_{1,3} = 0.85$, $PCC_{2,3} = 0.86$, a result consistent with the other periods described in Table 3.

The statistical analysis over 1772 hourly observations from the $30^{th}$ of May to the $14^{th}$ of December gives $PCC_{1,2} = 0.90$, $PCC_{1,3} = 0.95$ and $PCC_{2,3} = 0.93$ while $MAE_{1,2} = 4.4$ ppb, $MAE_{1,3} = 2.8$ ppb and $MAE_{2,3} = 4.4$ ppb.

### 3.4 Correlation and bias between low-cost sensors and reference. Inter-comparison

We measured the correlation and the bias between the LCSs and the reference instrument to evaluate the performances of the LCSs in a real case scenario where there could be no possibility to improve the laboratory calibration model. In addition to the calculation of the MAE we consider also the Root Mean Square Error (RMSE) and, to highlight the sign of the bias, the Mean Bias Error (MBE), defined as follows:

$$RMSE = \sqrt{\frac{\sum_{i=1}^{n}(e_i)^2}{n}} \tag{3}$$

| | Total | Jun | Jul | Aug | Sep | Oct | Nov | Dec |
|---|---|---|---|---|---|---|---|---|
| $PCC_{1,2}$ | 0.90 | 0.68 | 0.85 | 0.92 | 0.87 | 0.79 | 0.87* | 0.87 |
| $PCC_{1,3}$ | 0.95 | 0.87 | 0.95 | 0.96 | 0.92 | 0.62 | 0.85 | 0.84 |
| $PCC_{2,3}$ | 0.93 | 0.86 | 0.84 | 0.91 | 0.86 | 0.57 | 0.79* | 0.75 |
| $MAE_{1,2}$ | 4.39 | 6.69 | 5.72 | 3.35 | 3.30 | 4.14 | 3.48* | 4.50 |
| $MAE_{1,3}$ | 2.84 | 4.41 | 2.78 | 1.96 | 2.01 | 2.49 | 2.45 | 2.85 |
| $MAE_{2,3}$ | 4.44 | 3.60 | 5.12 | 3.50 | 3.24 | 4.89 | 4.87* | 7.00 |
| Obs. nr. | 1772 | 387 | 77 | 248 | 208 | 59 | 521 | 236 |

**Table 3.** Stability of LCSs intra-correlation (PCC) and bias (MAE) considering the period from the 30$^{th}$ of May to the 14$^{th}$ of December 2018. *Due to the malfunction of the Mrg2 sensor, 82 hourly observations from 2020-11-05 to 2020-11-08 of this sensor have been excluded from the LCSs dataset.

$$MBE = \frac{\sum_{i=1}^{n} e_i}{n} \quad (4)$$

In this contest the *true* value is the ozone measured by the reference instrument in Col Margherita. The statistical analysis, considering both the whole dataset and the trend of each sensor, is reported in Table 4. On average, the PCC between the sensors and the reference was $\approx 0.8$, with the smallest values registered in December. The average MAE was $\approx 5$ ppb and RMSE was $\approx 7$ ppb. The bias was not constant through the period. It was larger during summer and it decreased during autumn, with the MBE showing a change in the sign. Probable causes that might affect the accuracy of the LCS measurements could be the environmental temperature and relative humidity, whose dependence is also described in the sensor datasheet. We observed cases with poor agreement. Perhaps lower ozone concentrations and/or low environmental variation of ozone concentrations encountered in the mid winter periods influenced the data quality from the sensors or even the reference instrument. This could explain the low correlation observed during December. Also, as discussed in subsection 3.3, we hypothesized that the low correlation value between LCSs and reference during October could have been due to the short amount of period where both systems were running. The case of June for Mrg2 and for all the LCSs during July were peculiar. Perhaps a role played by the temperature difference between the inside and outside of the box could explain the lower correlation.

### 3.4.1 Relative humidity threshold

The Alphasense Application Note 106 (AAN 106) reports that the low-cost sensors must operate in the RH range from 15% to 90%. We evaluated that the exclusion of the observations collected outside the RH interval does not improve the correlation and accuracy metrics between the LCSs and the reference instrument. Thus, considering the poor improvement obtained through skimming, we did not exclude further LCSs measurements from the dataset.

Atmospheric Measurement Techniques
EGU
Discussions

| | | Total | Jun | Jul | Aug | Sep | Oct | Nov∗ | Dec |
|---|---|---|---|---|---|---|---|---|---|
| Mrg1 | PCC | 0.86 | 0.78 | 0.47 | 0.73 | 0.62 | 0.61 | 0.67 | 0.37 |
| | MAE | 4.83 | 5.92 | 9.01 | 6.30 | 4.47 | 2.73 | 3.45 | 3.69 |
| | RMSE | 6.32 | 7.21 | 11.27 | 8.08 | 5.76 | 3.63 | 4.35 | 4.74 |
| | MBE | 1.83 | 4.86 | 7.62 | 4.27 | 1.56 | 0.15 | -0.59 | -2.24 |
| Mrg2 | PCC | 0.79 | 0.36 | 0.46 | 0.75 | 0.70 | 0.65 | 0.64 | 0.36 |
| | MAE | 6.44 | 11.36 | 5.18 | 4.64 | 3.27 | 4.39 | 5.03 | 6.84 |
| | RMSE | 8.76 | 14.33 | 6.74 | 6.26 | 4.33 | 5.21 | 6.11 | 7.99 |
| | MBE | 0.76 | 10.75 | 2.70 | 2.01 | -0.63 | -3.92 | -3.58 | -6.71 |
| Mrg3 | PCC | 0.84 | 0.63 | 0.49 | 0.76 | 0.68 | 0.32 | 0.70 | 0.43 |
| | MAE | 5.27 | 9.00 | 8.07 | 6.27 | 4.19 | 3.77 | 3.06 | 3.00 |
| | RMSE | 7.13 | 10.74 | 10.42 | 7.89 | 5.32 | 4.76 | 3.97 | 3.68 |
| | MBE | 3.12 | 8.35 | 5.47 | 4.31 | 1.10 | 0.50 | 0.39 | 0.27 |
| | 1h-obs. nr. | 1772 | 387 | 77 | 248 | 208 | 59 | 521 | 236 |

**Table 4.** Inter-correlation and bias between the LCSs calibrated in the laboratory and the reference. The sum of the hourly observations in the monthly analysis differ with the total 1793 observations by the data from the $31^{th}$ of May to the $14^{th}$ of June.

### 3.5 Environmental low-cost sensors model

We evaluated the relationship between the bias and the temperature, relative humidity, atmospheric pressure, solar radiation, wind speed and wind direction to further investigate which meteorological variables were contributing to the bias ($e_i$) between the LCSs ($O_3^{lcs}$) and the reference ozone concentration ($O_3^R$). Figure 5 shows the correlation plots for temperature, relative humidity and LCSs signal while the remaining plots for the non correlating variables are reported in the Supplementary Figure S5.5. The bias showed a correlation trend with the air temperature (PCC $\approx 0.53$, p-value $< 2e-16$) and relative humidity (PCC $\approx 0.45$, p-value $< 2e-16$), while no evidence of correlation was shown with the incident solar radiation (PCC $\approx 0.05$, p-value $\approx 0.1$), the atmospheric pressure (PCC $\approx 0.24$, p-value $\approx 0.3$), the wind speed (PCC $\approx -0.22$, p-value $\approx 0.1$) or the wind direction (PCC $\approx -0.10$, p-value $\approx 0.4$). We finally observed that the bias was dependent on the sensor signal itself (PCC $\approx 0.55$, $p < 2e-16$).

We considered the following statistical multivariate linear model to evaluate the sensors bias under specific meteorological conditions and LCS signal, implementing in the model only the explanatory variables as previously described:

$$e_j = a_{0,j} + a_{1,j}T + a_{2,j}RH + a_{3,j}O_3^{lcs,j} \quad (5)$$

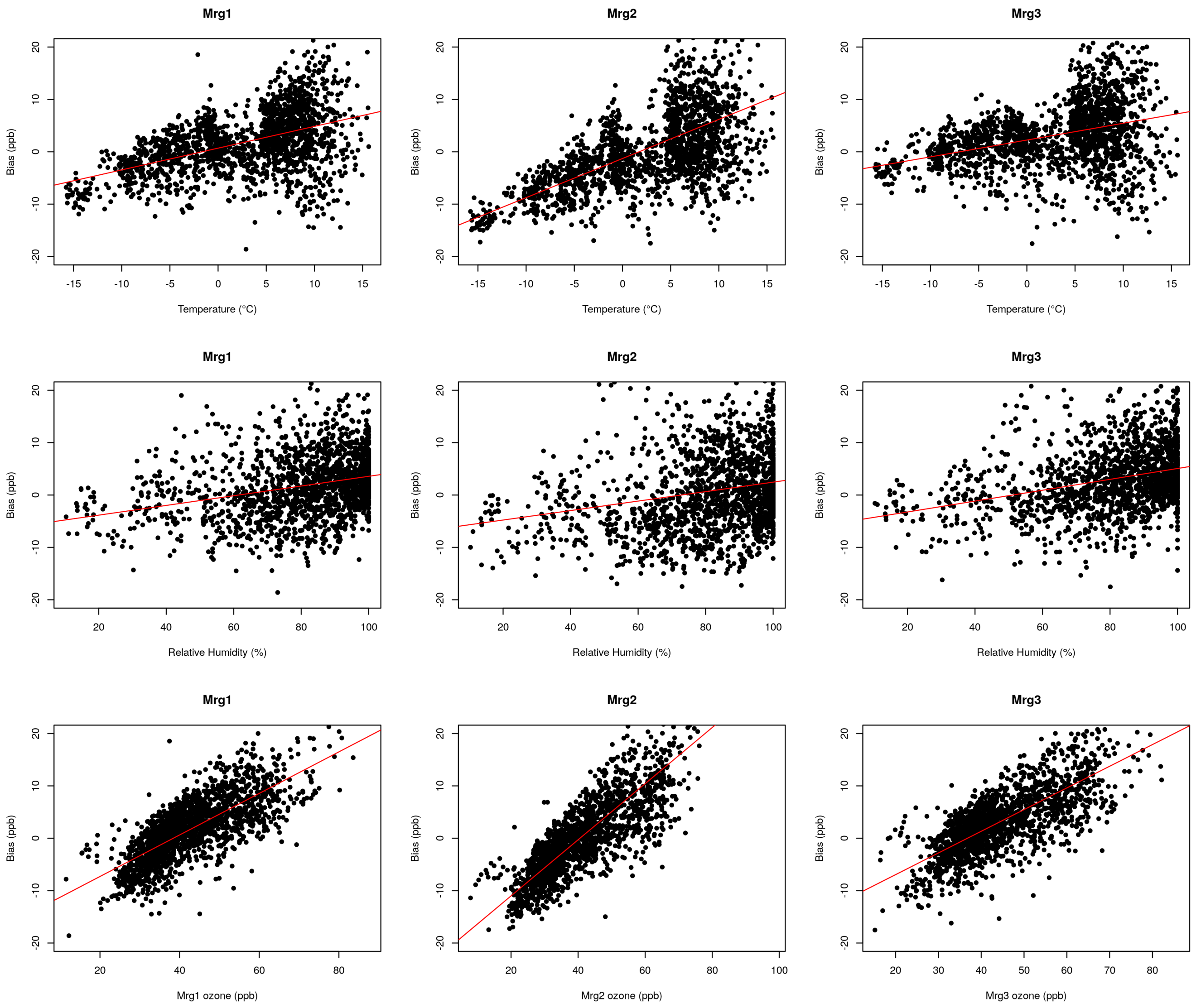

**Figure 5.** The environmental variables with a non negligible linear correlation with the LCS bias.

where bias ($e_j$, j=1,2,3) is the difference in the ozone concentration measurement between the j-esimo LCS and the reference, $a_{i,j}$ denote the model coefficients, $T$ is the ambient air temperature, $RH$ is the relative humidity and $O_3^{lcs,j}$ are the j-esimo LCS ozone reading obtained from Equation 1.

We thus used Equation 5 to improve the laboratory calibration model and to achieve a better estimation of the field (F) ozone concentration measured by the j-esimo low-cost sensors ($O_3^{F,j}$):

$$O_3^{F,j} = O_3^{lcs,j} - e_j \tag{6}$$

### 3.6 Performance of the model

We considered three case scenarios to evaluate the performance of the model. A summary of the values obtained are reported in Table 5.

#### 3.6.1 Scenario 1

The first scenario considered the whole reference dataset to model the bias and correct the LCSs measurements. The inter-correlation between the LCSs and the reference improved by $\approx 1\%$. The accuracy between the LCSs measurements and the reference improved by $\approx 60\%$ lowering the average bias (MAE) from $\approx 5.5$ to $\approx 3.2$ ppb, with the 50% of the bias distribution between $\approx \pm 2.5$ ppb and the 95% between $\approx \pm 8.5$ ppb (Figure 6). The corrected LCSs dataset, obtained by the continuous calibration of the LCSs in Col Margherita, could be used to reconstruct the environmental ozone concentration in case of loss of reference data, a situation that may occur due to power outages or during the instrumental calibration when the reference instrument is not present at the observatory. During our experiment we obtained 1556 1-hr additional ozone measurements, considering the time periods where the LCSs and the AWS were collecting data (Figure 7 and Supplementary Materials S5).

#### 3.6.2 Scenario 2

A second scenario still considered the whole reference dataset, but it aims to evaluate the intra-compatibility of the LCSs bias model. This might be useful if considering the use of the bias model of one low-cost sensor for the calibration of another low-cost sensor. This approach opens the possibility of performing a remote calibration in the surrounding area of the Col Margherita. A remote calibration allows the deployment of a local sensor network where remote standalone sensor's signal is corrected using the bias model studied in a location where a reference instrument is always present. We compared the coefficients of the bias model of each low-cost sensor. Subsequently, we evaluated the accuracy between one of the LCSs and the reference, correcting its dataset using the averaged model coefficients of all the three LCSs (scenario 2a). Next, we evaluated the accuracy between a LCS and the reference, correcting its LCS dataset using the averaged value of the other two LCSs coefficients (scenario 2b). The average value of the intercept coefficient ($a_0$) is -27.76 $\pm$ 8.94%. The average value of the temperature coefficient ($a_1$) is 0.26 $\pm$ 8.11%. The average value of the RH coefficient ($a_2$) is 0.11 $\pm$ 5.41% and the average values of the LCS coefficient ($a_3$) is 0.45 $\pm$ 11.32%.

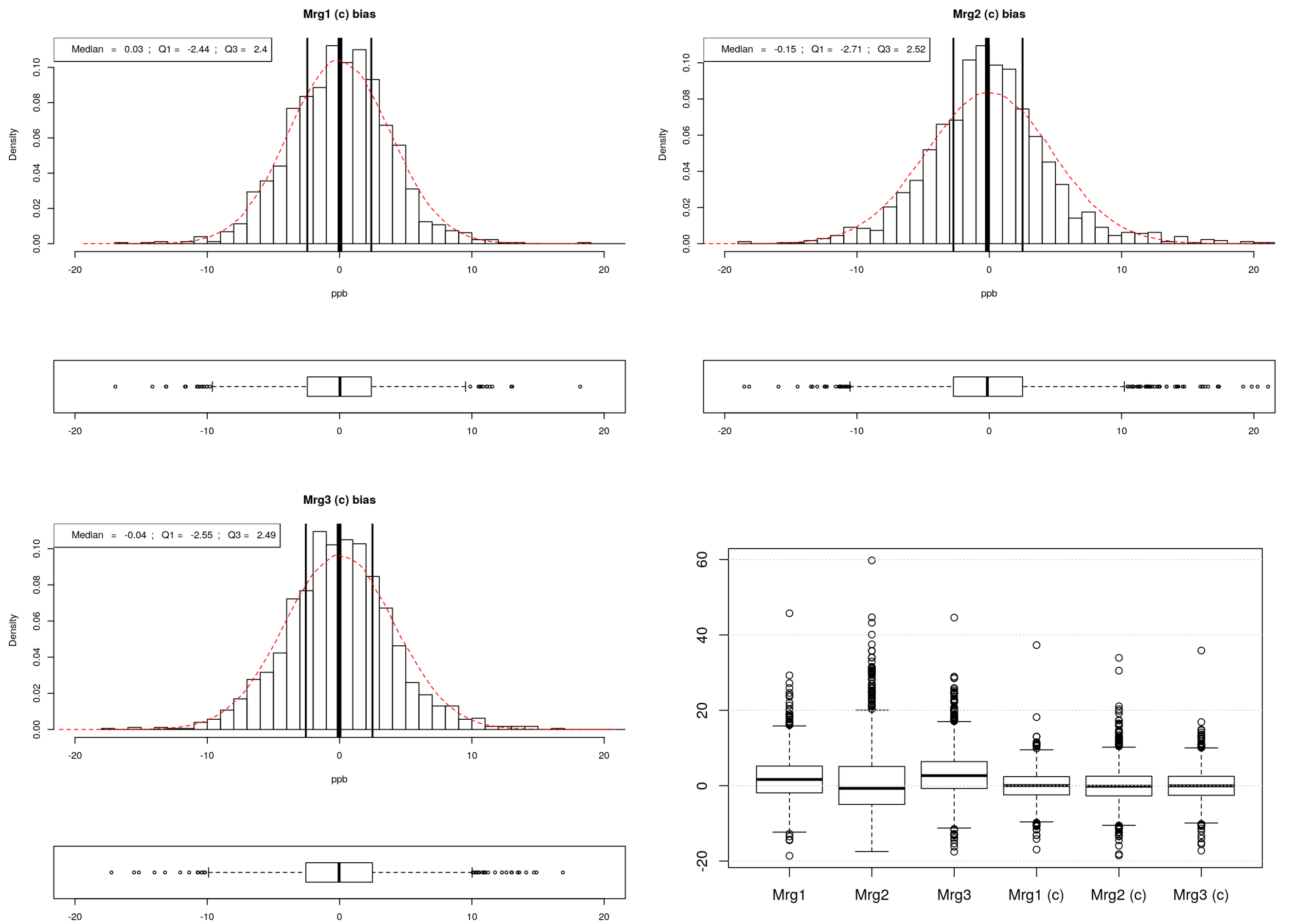

**Figure 6.** In order from top-left to bottom-right: (A) bias between the reference instrument and the Mrg1 sensor after the LCS correction, (B) bias of the Mrg2 sensor after LCS correction, (C) bias of the Mrg3 sensor after correction and (D) comparison of the LCSs bias before and after correction.

### 3.6.3 Scenario 2a

We considered the average coefficient values and their relative standard deviations to model the bias of each sensor and to calculate the corrected LCS ozone dataset. In this scenario the inter-correlation between the LCSs and the reference improved by $\approx 1\%$. The accuracy between the LCSs measurements and the reference improved, lowering the average bias (MAE) to $\approx 3.3$ ppb. The 50% of the bias distribution confidence interval (CI) was $\approx \pm 2.6$ ppb and the 95% CI was $\approx \pm 8.7$ ppb.

### 3.6.4 Scenario 2b

We corrected the ozone measurements of each low-cost sensor using the average model coefficients of the other two low-cost sensors. The inter-correlation metrics between the LCSs and the reference improved by $\approx 1\%$. The accuracy between the LCSs measurements and the reference improved, lowering the average bias (MAE) to $\approx 3.5$ ppb. The 50% of the bias distribution CI was $\approx \pm 3$ ppb and the 95% CI was $\approx \pm 9$ ppb.

### 3.6.5 Scenario 3

The third scenario examined the execution of consequential field calibrations. It represents the situation where there is no chance to lay on local or remote calibration. The low-cost sensing system has to be installed alone, except for scheduled calibration periods during which a reference instrument is placed aside the low-cost sensor to model the bias. The calibrating periods must be chosen to depict as much as possible all the seasonal meteorological conditions of the site.

We modeled the bias considering only the June and the December reference dataset. These periods represent the annual extremes of the meteorological conditions at Col Margherita (Table 2). Data used for field calibration are 423 1-hr observations from the $30^{th}$ of May to the $21^{st}$ of Jun 2018 and 236 1-hr observations from the $1^{st}$ to the $14^{th}$ of December 2018.

In June the ambient temperature ranged from 2 °C to 14 °C with a mean temperature of 7 ± 2 °C, ambient relative humidity (RH) ranged from 48% to 100% with a mean RH of 85% ± 10%, atmospheric pressure ranged from 747 hPa to 755 hPa with a mean pressure of 750 ± 2 hPa and wind speed ranged from 0.20 m/s to 12 m/s with a mean wind speed of 3.5 ± 2 m/s. The range of 1-hour validated $O_3$ concentration was 31.3 ppb with a maximum of 66.3 ppb measured by the Thermo 49c Ozone analyzer.

In December the ambient temperature ranged from -16°C to 3 °C with a mean temperature of -6.5 ± 5 °C, RH ranged from 38% to 100% with a mean RH of 75% ± 15%, atmospheric pressure ranged from 734 hPa to 751 hPa with a mean pressure of 742 ± 5 hPa and wind speed ranged from 1 m/s to 23 m/s with a mean wind speed of 6 ± 4 m/s. The range of 1-hour validated $O_3$ concentration was 26.6 ppb with a maximum of 43.1 ppb measured by the Thermo 49c Ozone analyzer.

In this scenario the inter-correlation between the LCSs and the reference improved by $\approx 1\%$. The accuracy between the LCSs measurements and the reference improved by lowering the average bias (MAE) to $\approx 3.3$ ppb. The 50% of the bias distribution CI was $\approx \pm 2.7$ ppb and the 95% CI was $\approx \pm 8.7$.

| | | Scenario 1 | Scenario 2a | Scenario 2b | Scenario 3 |
|---|---|---|---|---|---|
| Mrg1 | PCC | 0.88 | 0.88 | 0.87 | 0.87 |
| | MAE | 2.95 | 3.02 | 3.32 | 2.97 |
| | Median | 0.03 | 0.01 | 0.94 | 0.14 |
| | CI (95%) | -7.09 : 7.40 | -7.89 : 7.12 | -8.12 : 7.54 | -7.36 : 7.66 |
| Mrg2 | PCC | 0.80 | 0.80 | 0.80 | 0.79 |
| | MAE | 3.48 | 3.63 | 3.79 | 3.78 |
| | Median | -0.15 | -1.17 | -1.24 | -0.43 |
| | CI (95%) | -9.32 : 10.92 | -9.05 : 11.44 | -8.65 : 13.01 | -11.66 : 9.17 |
| Mrg3 | PCC | 0.85 | 0.85 | 0.85 | 0.85 |
| | MAE | 3.13 | 3.29 | 3.24 | 3.25 |
| | Median | -0.04 | 0.73 | 0.85 | -0.75 |
| | CI (95%) | -7.93 : 8.45 | -8.06 : 8.46 | -7.25 : 9.12 | -9.15 : 7.25 |

**Table 5.** Summary of the performaces of the LCSs bias model in the three scenarios considered.

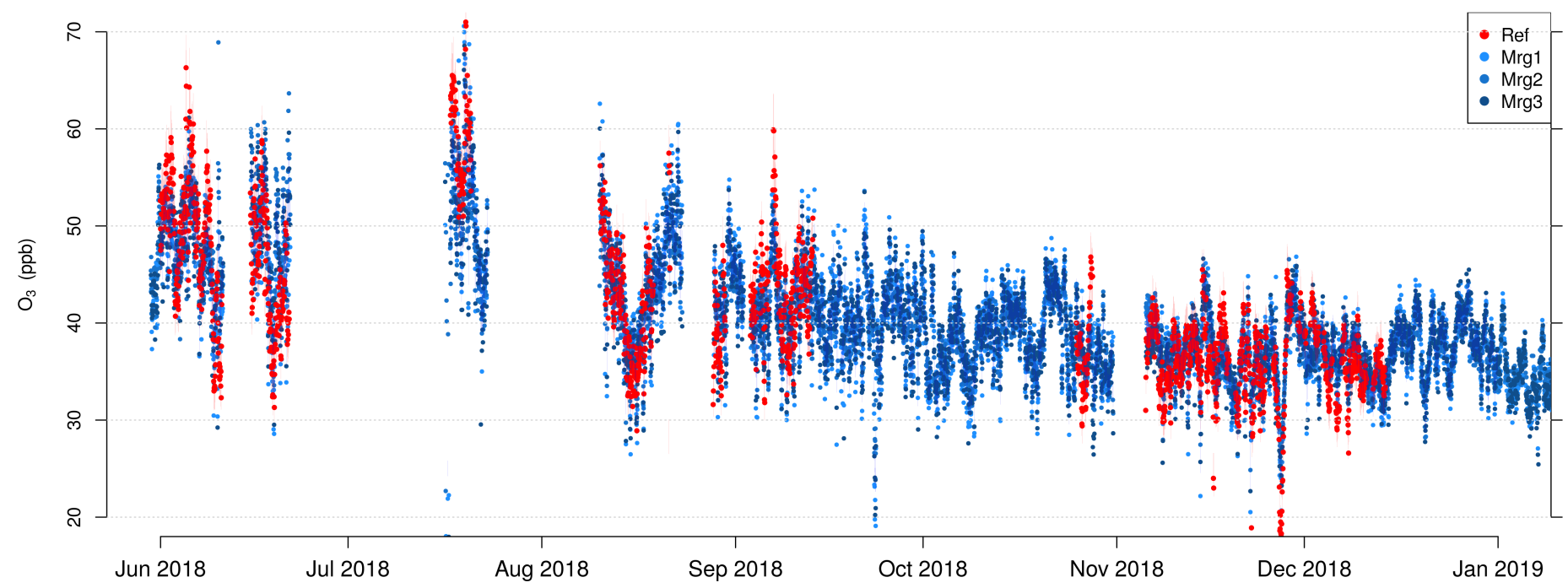

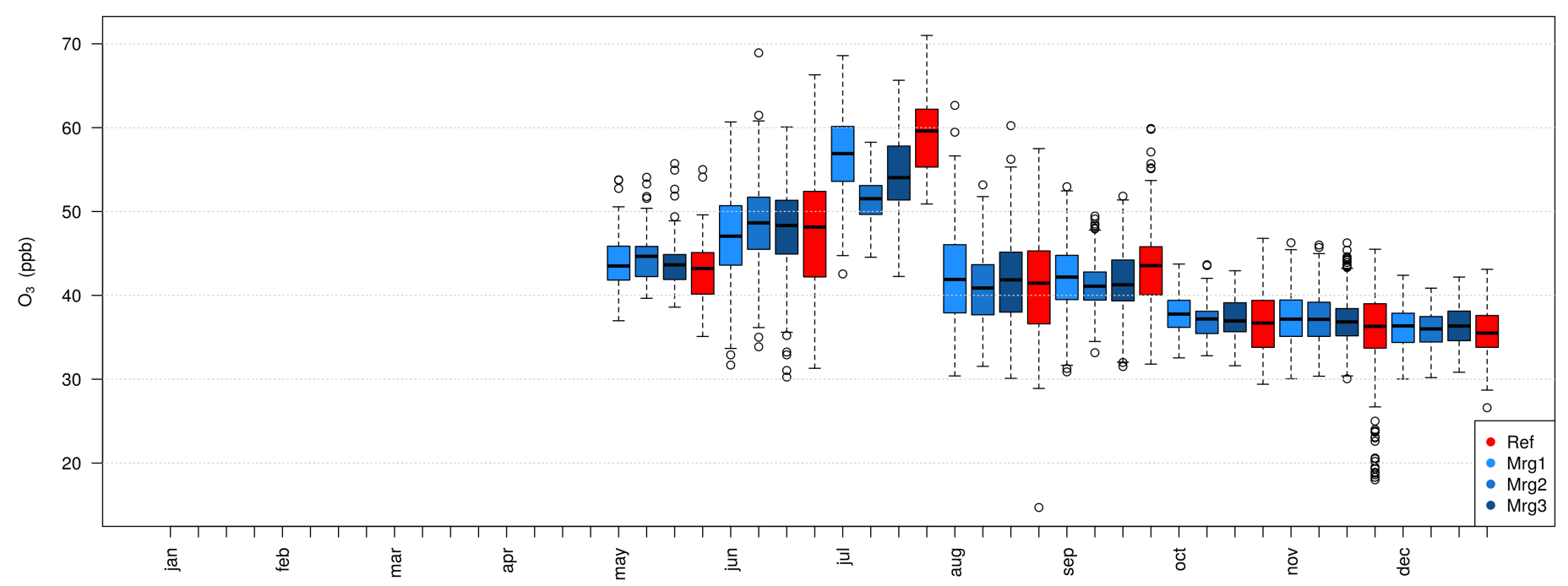

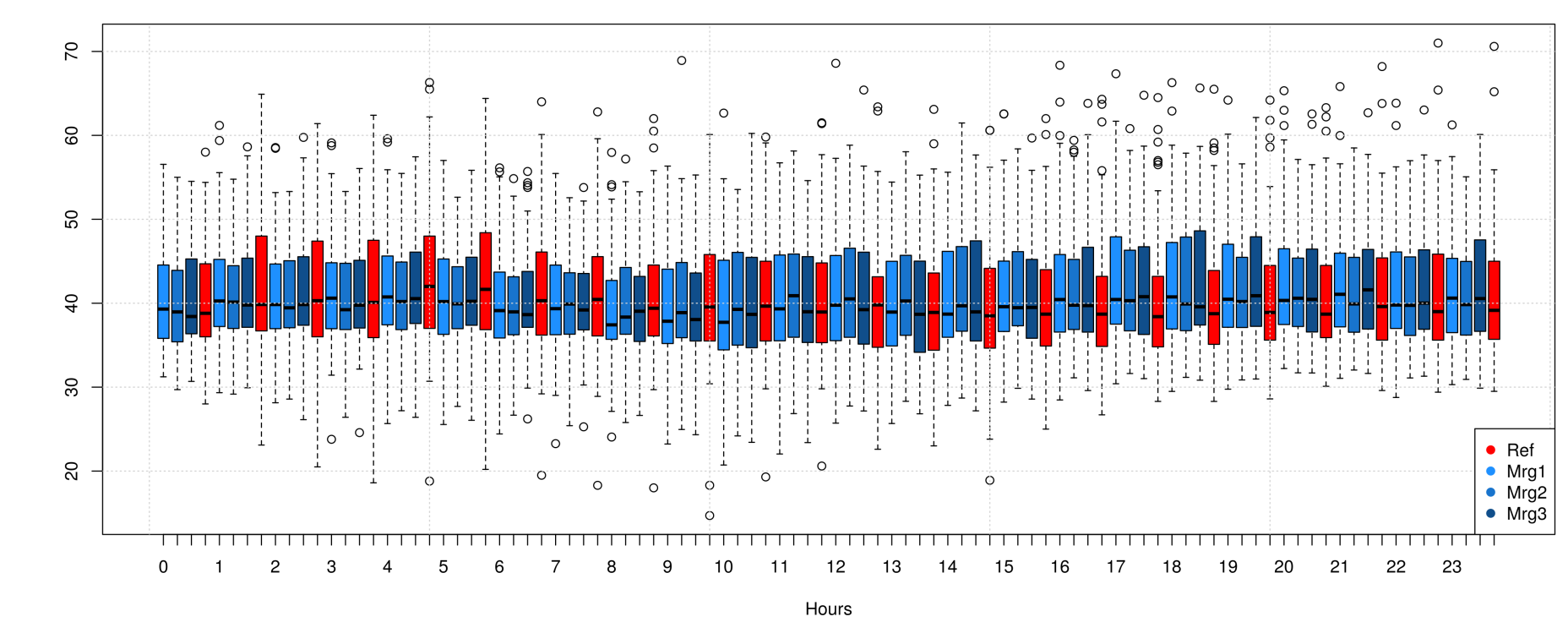

**Figure 7.** Time series of ozone concentration measurements and meteorological variables for the period June-December 2018. Monthly and hourly boxplots for the period June-December 2018. LCSs average values are reported in blue gradient and the reference ozone concentration is reported in red. Observations from periods where both the LCSs and the reference were operational have been considered for the monthly and hourly boxplots.

## 4 Discussion

We summarized the LCSs analytical performance results obtained through two laboratory calibration experiments and a $\approx 7$ months field experiment performed at the Col Margherita observatory.

### 4.1 Detection limits

The LCSs were capable of detecting ozone in the typical environmental $\mathrm{ppb}$ range. During the laboratory calibration experiments we observed that the LOD of the LCSs was about 5 $\mathrm{ppb}$ and the LOQ was about 15 $\mathrm{ppb}$. We noticed that the LCSs response to ozone concentration was linear ($PCC > 0.99$) up to 250 $\mathrm{ppb}$ and these analytical performances of the LCSs did not change after two laboratory calibration experiments, conducted three months apart one from another. When used in the field, the linear regression coefficients obtained through laboratory calibration, at fixed temperature and humidity, are not capable to fully describe the behavior of the LCSs, $PCC \approx 0.8$.

### 4.2 LCSs calibration

We observed that all the sensors require an individual, both in laboratory and in the field, calibration and caution should be used if it is decided to rely on the declared calibration values, which could be insufficiently accurate for an environmental study, even if variation in ozone concentrations are clearly detectable. Careful should be used if deciding to perform a single field calibration since this can be insufficient as it is very unlikely that, at mid-latitudes, one could choose a representative period of the whole year. A solution might be to perform many field calibrations, covering at least the extremes of the meteorological year condition, in order to depict as much as possible the behavior of the sensors in the environment. It may be considered to perform a more exhaustive laboratory calibration in a climatic chamber simulating the environmental conditions where the sensor will be deployed, to improve the understanding of the LCSs.

### 4.3 Precision

The evaluation of the LCSs intra-correlation during the field experiment revealed that the sensors behave in a similar but not identical way ($intra - PCC \approx 0.9$), thus it is important to increase the reliability and reproducibility of the measurements considering LCSs redundancy. This is particularly important when using the sensors in the field without any reference. We detected indeed a case during which one of the three LCSs was not working properly (Supplementary Figure S2.3). These anomalies could not be detectable if redundancy is not considered. The precision of the LCSs, described by the RSD, was evaluated in the laboratory with a constant reference ozone concentration. We observed that near the LOD the LCSs RSD is $\approx 3.5\%$, decreasing to $\approx 1.6\%$ at $\approx 50$ $\mathrm{ppb}$ and reach the asymptotic value of $\approx 0.8\%$ for larger ozone concentrations (> 200 $\mathrm{ppb}$). We did not notice a deterioration of the LCSs precision between the two laboratory calibration experiments. During the field campaign it is not trivial to evaluate the precision in terms of RSD, because there is neither a constant reference ozone concentration nor a constant meteorological situation. Thus, since it is not possible to evaluate the precision in field, we assumed that the precision during the field experiments was compatible with the precision studied in the laboratory.

### 4.4 Bias

The LCSs average bias, measured through the MAE between the LCS and the reference instrument during the field experiment, was $\approx 3.5$ ppb ($\approx$ 6 ppb without performing the bias correction). The LCSs electrical noise measured in the laboratory was $\approx 1.5$ ppb (see Supplementary Figure S3.1) and bias > 4 ppb. During the field experiment we observed that the LCSs bias is dependent from the air temperature, RH and from the LCS electrodes voltage. It is possible to build a multivariate linear model able to describe about 60% the bias variance during the period in which the LCSs operates aside the reference instrument. It is of interest that the environmental model coefficients do not differ much one from another ($\approx$ 8.5%). We noticed that it is possible to reduce the bias of one LCS using the average coefficient values of another LCS, suggesting the development of a remote field calibration of the sensors in the surrounding area of the observatory.

### 4.5 Reliability

We have not measured differences in the analytical performances between the two LCSs calibration experiments. We showed that during the field experiment the LCSs response to the reference ozone concentration was affected by the local meteorology and their dependence can be analytically described. We observed that a time dependent model for correcting the bias was not necessary for our field experiment, but would be reasonably needed for long-term monitoring, when LCSs aging will not be negligible. The rapid response ($\approx 30$ ppb $\cdot$ s$^{-1}$) and the small memory effect, measured both during the laboratory experiments and in the field experiment, allowed the measurement of rapid variations of the ozone concentration.

The sensor's datasheet reports that $NO_2$ interferes with the ozone measurements. Since at the MRG observatory there was not a $NO_2$ analyzer during our field experiment, we were not able to measure the concentration of this gaseous pollutant.

We referred to a local study(Costantino, 2016) to deduce the amount of $NO_2$ that could have interfered with our experiment. This work was based on the data of the regional environmental agency: a survey on ozone and nitrogen dioxide at Passo Valles (2.032 m a.s.l., about 3.2 km away from the MRG and about 500 m of lower altitude than MRG) was conducted, from the $1^{st}$ of January 2007 to the $1^{st}$ of January 2011. The average annual concentration of $NO_2$ at Passo Valles was $4 \pm 1$ ppb and the average annual concentration of ozone was $\approx 94 \pm 5$ ppb. As a comparison, despite referring to different years, the ozone concentration at Col Margherita was $\approx 40 \pm 10$ ppb during 2018, $\approx 57\%$ lower than what was recorded at Passo Valles. It is worth to notice that the monitoring site at Passo Valles is next to a road, which could be a local source of $NO_2$ pollution. Therefore, even if it was not possible to know the $NO_2$ concentration at MRG, the mean values measured at Passo Valles suggest that a few ppb of $NO_2$ may have interfered with our experiment in MRG. Nevertheless, since it has been observed that the $NO_2$ concentrations are decreasing all over Europe(Jamali et al., 2020; Castellanos and Boersma, 2012), it is unlikely that $NO_2$ interference could have been detected by our LCSs system in MRG and thus explain some of the bias we observed between the LCSs and the reference instrument.

We showed that the LCSs was subjected to many considerable "stresses" throughout the field experiment. During the summer period we faced some severe power outages which caused data losses. These events were due to heavy thunderstorms and bad weather conditions that are characteristic of the alpine summer season. Moreover, the LCSs faced a severe storm ("Vaia",

$29^{th}$ October 2018, see Supplementary Figure S5.1) which caused several damages all through the North of Italy and caused a general black out in the Col Margherita area. It is worth noticing that the LCSs system was the only one that remained on during the Vaia storm blackout. When the power returned, the system showed no damage.

## 5 Conclusions

We found that $O_3$ low-cost gas sensors can provide concentration measurements with a bias of few ppb ($\pm 8.5$ ppb at 95% of confidence) throughout the period of the field operation.

We remarked that all the sensors require an individual calibration. We observed that the laboratory calibration is not sufficient to explain the behavior of the sensors during the field experiment. As a consequence, performing a sensor field calibration near a reference site is necessary and this requires infrastructure. Since the quality of the sensor calibration depends on the description of the environmental conditions (i.e. pollutants concentrations, meteorology), we showed that the reference instrumentation is relevant to perform periodic field calibrations.

In this work we discussed three procedures for the LCSs field calibration which can be used for the design of a low-cost sensors network in an alpine region. We showed how to improve the LCSs analytical performances when the reference instrument is always present, when it is present only for scheduled calibration periods and finally how to improve the $O_3$ measurements of a standalone LCSs using a remote calibration.

The comparison with a reference grade instrument revealed that the sensor's bias is impacted by changes in environmental temperature and relative humidity. These effects are demonstrated to be reduced by the application of a correction function and we revealed that a multivariate linear model is capable of describing up to 60% of the bias variability. We noticed that the bias model coefficients were comparable between each sensor ($\approx$ 8.5 % of difference). This opens the chance to perform a remote calibration of low-cost sensors which are not in the proximity of the reference instrument. Future studies could focus on the improvement of the mathematical description of the LCSs working principle and on its environmental dependence, in order to evaluate to which extent a single bias model could be used in a sensors network in an alpine area.

Achievable technical improvements for the enhancement of the analytical performances of the LCSs system are still open as the low-cost technology is getting better. Considering an improved thermal insulation of the sensing system, while ensuring the circulation of the ambient air without impacting the energy efficiency of the instrument, might be taken into consideration for future works. This study reveals how to obtain valuable data from a low-cost instrument in a remote harsh alpine environment and shows procedures for the design of adequate monitoring strategies in the study of tropospheric gases in remote areas.

*Data availability.* The data used in this study can be obtained from the authors upon request.

*Author contributions.* Federico Dallo: Conceptualization, Data curation, Formal analysis, Methodology, Software, Writing - original draft, Writing - review & editing. Daniele Zannoni: Data curation, Formal analysis, Writing - review & editing. Jacopo Gabrieli: Conceptualization, Methodology, Writing - review & editing. Paolo Cristofanelli: Conceptualization, Writing - review & editing. Francescopiero Calzolari: Data curation. Fabrizio de Blasi: Data curation, Formal analysis, Writing - review & editing. Andrea Spolaor: Data curation, Writing - review & editing. Dario Battistel: Data curation, Writing - review & editing. Rachele Lodi: Data curation, Writing - review & editing. Warren R.L. Cairns: Writing - review & editing. Ann Mari Fjæraa: Conceptualization, Funding acquisition, Writing - review & editing. Paolo Bonasoni: Funding acquisition, Writing - review & editing. Carlo Barbante: Funding acquisition, Writing - review & editing.

*Competing interests.* The authors declare that they have no other competing interests.

*Acknowledgements.* We thank the European Commission for funding GMOS, as part of the FP7 (Contract no. 26511), during which the MRG observatory was built. The measurements of this study were supported by ERA-PLANET (www.era-planet.eu) and the trans-national project iGOSP - Integrated Global Observing Systems for Persistent Pollutants (www.igosp.eu), funded under the EU Horizon 2020-SC5-15-2015 "Strengthening the European Research Area in the domain of Earth Observation", type of action: ERA-NET-Cofund Grant, (Grant Agreement N. 689443). We thank for the financial support given by the National Project of Interest Next-Data (http://www.nextdataproject.it/) by the Italian Ministry for Education, University and Research (MIUR), for the ozone and meteorological measurements. This work was part of the O3NET project, funded by the Research Council of Norway through an Arctic Field Grant (ES607473, Research in Svalbard ID: 10940). This project has received funding from the European Union's Horizon 2020 research and innovation programme under the Marie Skłodowska-Curie grant agreement No. 844526. We kindly thank Enrico Natin and Roberto Epis, form the Workshop of the Ca' Foscari University, and Roberto Marin, from the Area Servizi Informatici e Telecomunicazioni of the Ca' Foscari University for their technical support. We thank Meteotrentino for providing the solar radiation and precipitation data from the weather station of the Passo Valles (Trento, Italy) (https://www.meteotrentino.it). Finally, a special thank to Renzo Minella, Loris Scola and the "Ski area San Pellegrino" people (https://www.ski area sanpellegrino.it) for their fundamental cooperation and support during the field activities at the Col Margherita Observatory.

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
