# Peer review of "Calibration and assessment of electrochemical low-cost sensors in remote alpine harsh environments"

_Atmospheric Measurement Techniques, 2020_

## Author Response (AR1)

Replies to Anonymous Referee #3

**Referee #3: "The manuscript is a little wordy and is large in page scale."**
Reply: To reduce the page number we moved some of the main text to the Supplementary Materials.
• Lines from 87-91 are moved to the Supplementary Material.
• Lines from 95-101 are moved to the Supplementary Material.
• Lines 115-120 and 124-126 are moved from 2.4 to 2.3. Lines 121-124 are moved in Supplementary Material and 2.4 has been eliminated from the manuscript.
• Lines 207-214 are moved to the Supplementary Material and we rephrased the lines between 199-206 to maintain the description of sensor Mrg1 as we believe this is a useful resource for the reader.
• We moved figure 7 to the Supplementary Material.

**Referee #3: "All tables are too simple to be understood by the reader and more detailed information have to be added."**
Reply: We agree that we can improve the tables readability by adding more explanatory text in each table caption.
• Captions of Table 3,4 and 5 have been expanded.

**Referee #3: "In figure 1 it is better to show the sampling site in a larger scale of map in bottom panel."**
Reply: We modified the map accordingly.

**Referee #3: "The section of 4.2 can be deleted from the paper since it is unnecessary information."**
Reply: In this case, considering the importance of the laboratory and the field calibration of the LCSs, we do not fully agree with the Referee #3. We are more in agreement with Referee #1 who states, in their report on 10 Feb 2021, that "*Section 4.2 discussion on LCS calibrations is a good start as a lesson learned for others.*" and with Referee #4 comment who asked for further discussion on this topic. We nevertheless agree with Referee #3 that this paragraph can be improved and simplified so we rephrased and simplified line 345 and deleted the last sentence (lines 348-350) that contained unnecessary information.

**Referee #3: "The section of 3.1 Laboratory calibration of LCSs can be more concise since the section is unimportant in the work. The authors can be pay more attention on the field employment."**
Reply: We agree with Referee #3 that for the purpose of this work the field employment is more important than the laboratory calibration of the LCSs and we have shortened the discussion in 3.1 as described above.

**Referee #3: "In section 4.2 line 345, this sentence is complicated to reader and can be simpler."**
Reply: We rephrased and simplified lines 343-345.

Replies to Anonymous Referee #4

**Referee #4: "Highlight that the selected sensor are actually sensitive to NO2 (also) …"**
Reply: We added in the main text (section 2.3) a specific indication that the OX-B4 are sensitive to NO2.

**Referee #4: "The description of the auxiliary electrode working principle is imprecise."**
Reply: We thanks the Referee to point out this. In the Material and Methods we were referring to the Alphasense data-sheet. We changed the text of section 2.3 accordingly to the Referee indications and we moved lines 95-101 to the Supplementary Material.

**Referee #4: "Discuss the cost and the long term accuracy of the field calibration. Is the relative stability of the alpine environment beneficial for field calibration robustness?"**
Reply: We thank Referee #4 for this interesting question. We included a short paragraph in section 4.2 of the Discussion to briefly discuss this topic that will be further investigated in future work.

**Referee #4: "Calibration transfer 'absolute' results."**
Reply: This is a very interesting aspect of the work, and we are glad that the referee paid attention to this. The Referee is right, we are considering different case scenarios that simulate the possibility of remotely transferring the calibration parameters between LCSs, but with the assumption that the dependence on the environmental condition would not have change if we have _theoretically_ moved a sensor away from the Col Margherita Observatory. This assumption, that arises in section 3.6.2 lines 293-295, could not be true and will be matter of further investigation. Since this is a very important aspect of the work, we expanded the discussion in 3.6.2 and 4.2 following the indication of Referee #4.